# Modified Electrospun Membranes Using Different Nanomaterials for Membrane Distillation

**DOI:** 10.3390/membranes13030338

**Published:** 2023-03-14

**Authors:** Muzamil Khatri, Lijo Francis, Nidal Hilal

**Affiliations:** NYUAD Water Research Center, New York University Abu Dhabi, Abu Dhabi P.O. Box 129188, United Arab Emirates; mk8936@nyu.edu (M.K.); lf2426@nyu.edu (L.F.)

**Keywords:** electrospinning, nanofibers, nanomaterials, membrane separations, desalination, surface modification

## Abstract

Obtaining fresh drinking water is a challenge directly related to the change in agricultural, industrial, and societal demands and pressure. Therefore, the sustainable treatment of saline water to get clean water is a major requirement for human survival. In this review, we have detailed the use of electrospun nanofiber-based membranes (ENMs) for water reclamation improvements with respect to physical and chemical modifications. Although membrane distillation (MD) has been considered a low-cost water reclamation process, especially with the availability of low-grade waste heat sources, significant improvements are still required in terms of preparing efficient membranes with enhanced water flux, anti-fouling, and anti-scaling characteristics. In particular, different types of nanomaterials have been explored as guest molecules for electrospinning with different polymers. Nanomaterials such as metallic organic frameworks (MOFs), zeolites, dioxides, carbon nanotubes (CNTs), etc., have opened unprecedented perspectives for the implementation of the MD process. The integration of nanofillers gives appropriate characteristics to the MD membranes by changing their chemical and physical properties, which significantly enhances energy efficiency without impacting the economic costs. Here, we provide a comprehensive overview of the state-of-the-art status, the opportunities, open challenges, and pitfalls of the emerging field of modified ENMs using different nanomaterials for desalination applications.

## 1. Introduction

Water shortages have emerged as one of the century’s primary concerns because of industrialization, climate change, rising population, and modernization. Water is regarded as a basic requirement and source of sustenance for all living species on the planet; however, waste such as industrial effluents, heavy metals, volatile organic compounds, oil emulsions, and such impurities are major threats to marine life [1], which ultimately increases the demand for the development and improvement of water remediation technologies concerning recyclability and sustainability [2,3,4,5,6]. Amongst various water treatment techniques, membrane distillation (MD) is considered one of the leading-edge technologies for obtaining drinkable water from seawater, brine, or other wastewater resources [7]. The MD process can be operated with renewable energy sources such as solar, geothermal, or other waste-heat energy sources for low-cost water treatment [1,8]. However, many improvements and optimizations are required for their efficient production and deployment on a large scale [9,10,11,12,13,14,15].

In simple words, in an MD process, liquid–vapor separation occurs at the interface of a highly hydrophobic microporous membrane. A trans-membrane vapor pressure difference created due to the temperature difference across the membrane is the driving force in an MD process. Theoretically, the hydrophobic layer has the potential to reject non-volatile pollutants up to 100% that may be dissolved in the feed [16]. As a result, MD has received significant attention in water recovery from saline water as well as wastewater. However, when dealing with multiple effluents, including various types of low surface-tension components such as oils, grease, alcohols, organics, and surfactants, the membrane’s hydrophobicity becomes a concern, due to the affinity of various pollutants to the membrane surface. In this scenario, a pretreatment process is recommended before the MD process to remove all the organic pollutants [17].

Many approaches have been made to further improve the polymeric membranes for water treatment [9,18,19]. Various forms of membranes for water reclamation have been reported, including nanofibers [3,16], foams [20], films [21], and other suitable textures, depending on the selectivity or overall water filtration requirements [22,23]. Because of their three-dimensional structure, high porosity, easy processing, and easy modification, electrospun nanofiber-based membranes (ENMs) have been widely prepared and optimized for water treatments, with consistent improvements in filtration/distillation/adsorption applications [9,24,25,26]. ENMs have been fabricated and optimized for MD process applications in many ways to meet the desired characteristics, which include high porosity, high permeability, low fouling/scaling propensities, low thermal conductivity, high liquid entry pressure (LEP), high hydrophobicity or increased water contact angle (WCA), and considerable mechanical strength [27,28,29,30].

Commercial membranes used for microfiltration are based on hydrophobic polymers such as polypropylene (PP), polytetrafluoroethylene (PTFE), and polyvinylidene fluoride (PVDF). These are fabricated via a melting technique followed by stretching or phase inversion and were utilized in the initial stages of the lab-scale MD testing. ENMs produced via electrospinning have remarkable characteristics to be utilized for MD because in the electrospinning process, users have more control over pore size, porosity, fiber size, and ultimate effective surface area. It also offers tunability of the surface morphology, affecting roughness as one of the important properties beneficial for the MD process, because the roughness enhances the hydrophobicity on the surface of ENMs.

However, to mitigate the pore-wetting and other related hindrances such as fouling and scaling during the MD process, the surface of ENMs has been modified using different available active nanomaterials, such as organic frameworks, zeolites, oxides, and other carbon-based nanomaterials [31]. This minireview gives a detailed insight into the new developments in electrospinning, and recent efforts to modify the ENMs using different nanomaterials to enhance the MD membrane characteristics and the MD process [32,33,34].

## 2. Conventional MD Configurations

Conventionally, four distinct MD configurations are used, namely, direct contact membrane distillation (DCMD), air gap membrane distillation (AGMD), sweeping gas membrane distillation (SGMD), and vacuum membrane distillation (VMD), which are developed based on the design of the coolant side of the membrane. In all the conventional configurations, the hot feed stream is in direct contact with the selective hydrophobic layer of the membrane. Figure 1 demonstrates the schematic representation of conventional MD module configurations. DCMD is the mostly used design for lab-scale investigations, in which the coolant stream is in direct contact with the membrane surface. For the AGMD process, an air gap is fixed between the coolant and the membrane, whereas, in SGMD, the permeate vapors coming from the hot-feed side through the membrane pores are collected with an inert gas that sweeps the H_2_O vapors and condense outside the module [35]. On the other hand, in the VMD process, a vacuum is applied on the permeate chamber of the membrane module which collects the permeate vapors and condenses them outside the membrane module. In DCMD and AGMD, the water permeate condensation process takes place inside the membrane module. Recent studies have reported new designs for MD process configurations, including submerged membrane distillation (SMD) [36,37], vacuumed air-gap membrane distillation (V-AGMD) [36], liquid or permeate gap membrane distillation (L/PGMD) [38], conductive gap membrane distillation (CGMD) [38], material gap membrane distillation (MGMD) [39], and flashed-feed vacuum membrane distillation (FF-VMD) [36].

## 3. New Developments in Nanofiber-Based Membranes

A variety of ENMs have been utilized for various sectors including water treatment [22,23], biomedical [41], energy [8,42], textile [43,44,45], cosmetics [46], aerospace [47], and environment [9,25] applications. To satisfy the huge demand for nanofibers, many scientists have been trying to increase the rate of production of ENMs to be practically utilized on an industrial scale, specifically for water reclamation applications [9,25,48,49]. Many techniques, including bubble electrospinning [50], needleless electrospinning [51], multi-jet electrospinning [52], electrospinning writing [53,54], centrifugal electrospinning [3], deep eutectic solvent (DES) electrospinning [24], and near-field electrospinning, [55] have been employed to produce ENMs.

Various functional polymers and guest molecules are available for fabricating ENMs, and their selection for fabricating membranes is based on the end purpose [11,27]. The literature suggests a big research gap in the precise optimization of efficient MD membrane fabrication. The incorporation of functional nanomaterials or simple optimized blending of two or more polymers can sometimes fulfill some of the requirements of the MD process, but it will require intensive optimization of electrospinning parameters to prepare ENMs with desired characteristics [27,56,57,58,59,60]. The surface morphology of membranes is a very important factor for water treatment applications [61,62,63], which can be tuned by taking all electrospinning parameters and functional properties of nanomaterials into consideration [64].

Figure 2a,b depict the growing interest of researchers, who have developed novel ENMs for the MD process in terms of patents and peer-reviewed research articles, respectively. This increasing research trend suggests the need for optimized and efficient ENMs for MD application. Moreover, substantial improvements such as large-scale production and reducing the wastage during an electrospinning process, along with the automation of the process, have to be addressed. The effect of electrospinning parameters on the electrospinning process and the characteristics of resultant ENMs are given in Table 1.

According to the literature, pristine polymeric ENMs cannot reach the required process performance and efficacy, thus limiting their practicability [18,73,74]. Various approaches have been used to enhance the hydrophobicity, permeate flux, and mechanical strength of the ENMs for the MD process [75]. Post-heat-treatment has been performed to enhance membrane characteristics such as mechanical strength, narrow pore size distribution, and optimized pore size [59,76].

Apart from the commonly used PVDF or polyvinylidenefluoride-co-hexafluoropropylene (PVDF-HFP), hydrophobicity of the MD membranes and the MD process performance were further improved when researchers fabricated ENMs with in-house synthesized polytriazoles and aromatic fluorinated polyoxadiazoles. Thus, the lowest possible surface energy of the base polymeric material has a very important role in maintaining and enhancing the hydrophobicity, surface roughness, WCA, and LEP of an ENM [29,34,77].

The incorporation of nano-additives into ENMs would further tune membrane surface characteristics and the MD process performance [78]. Modification of electrospun MD membranes using functional nanomaterials such as metal–organic frameworks (MOFs) [79,80], silica (SiO_2_), titania (TiO_2_), zeolites, carbon-based nanomaterials including CNT [81], graphene oxide (GO) [82], and activated carbon (AC) [83] are detailed in the following section.

## 4. Modification of ENMs with Functional Molecules and Their Effect on the MD Process

### 4.1. Effect of MOFs and Zeolites on ENMs and the MD Process

#### 4.1.1. Aluminum Fumarate (AlFu) Addition

MOF-based ENMs are being widely explored in water treatment applications. MOFs possess unique characteristics such as high porosity and high specific surface area, and they can be functionalized with a variety of nanomaterials so that we can tune the MOF-based membranes as per the need of the MD process application. Leaching of nanomaterials from the polymer matrix is one of the problems faced during the water filtration process. This may lead to contamination of the product water with the nano-sized MOFs. The electrospinning of MOF-based polymer dope solutions followed by pyrolysis under controlled conditions has the potential to resolve the leaching issues [79], physically trapping them into the nanofibers, which ultimately resolves the issues of chemical compatibility as well the mechanical stability of the resultant membranes [79,80].

The current research trend on MD membrane fabrication reveals that the optimum incorporation of nano-additives significantly improves the performance of the MD process by improving the aforementioned MD membrane characteristics [84]. AlFu is a commercially available and widely used MOF for MD membrane fabrication. AlFu-based MOF is inexpensive, and it comes with multiple MD-friendly characteristics, including being hydrophobic, water stable, environmentally friendly, able to be sourced from water and simple aluminum salts, having a permanent three-dimensional highly porous structure and with a large-scale production capacity about 3600 kg/m^3^ per day. Therefore, an AlFu MOF is considered a promising agent for wastewater treatment and possibly other MD applications when utilized with ENMs [85]. AlFu can be incorporated into PVDF-HFP via electrospinning. The observed permeate flux was improved up to 114% during 46 h of continuous MD operation in comparison with neat PVDF-HFP ENMs, as shown in Table 2. The thermal efficiency during the DCMD process was improved with the addition of AlFu MOFs in PVDF-HFP ENMs. The presence of AlFu MOF nanomaterials also enhanced the surface roughness, WCA, LEP, and thereby anti-wetting characteristics of the MD membrane [74]. Figure 3a shows a schematic diagram of the DCMD process while using AlFu-MOF-incorporated ENMs, revealing systematically ordered pores in MOFs with sizes between 0.6 and 0.7 nm [80], which offer an additional path for vapor transport and result in enhanced permeate flux [85]. Moreover, the presence of AlFu MOFs on the surface of ENMs in the form of protrusions enhances the surface roughness, which effectively increases the area for evaporation [64]. MAF-4 is a hydrophobic nanomaterial and interlayer that can be fabricated by seeding Zn (II) using the in-situ crystallization method. Because of the hydrophobic and anti-fouling characteristics of MAF-4, it has great potential to be utilized for MD. Figure 3b shows the step-by-step growth of MAF on a poly ether sulfone substrate via self-polymerization of Zn-seeded dopamine [86,87].

Only a few reports have been published on the application of MOF/MAF nanomaterials in MD membranes. In a recent report, 8% MAF-4 with PVDF could enhance water flux by 60% compared with neat PVDF membrane [88].

#### 4.1.2. Zeolitic Imidazolate Frameworks (ZIFs) Addition

The surface roughness and hydrophobicity of ENMs can be also enhanced by the addition of zeolites. A schematic representation of ZIF structures is given in Figure 3c. The zeolites belong to the MOF family that connects their imidazolate group with divalent metal cations [89]. ZIFs have high thermal resistance, chemical stability, and high porosity. The aforementioned characteristics of ZIFs can be utilized for applications including the fabrication of efficient MD membranes for water reclamation [90,91]. ZIFs contain organophilic imidazolate linkers, and these offer hydrophobic characteristics [92]. Additionally, Zeolitic imidazolate framework-71 (ZIF-71) offers enhanced hydrophobic properties due to the presence of methyl (–CH_3_) and chlorine (–Cl) entities in their chemical composition [89].

ZIF-71 NPs incorporated into PVDF-HFP significantly increased the surface roughness and hydrophobicity, yielding a permeate flux of 19.2 L m^−2^ h^−1^ with a salt rejection of >99.99%, which is 284% and 949% higher compared with the MD flux observed while using various pristine microporous membranes [91]. The PVDF-HFP ENM only had a WCA of 127.6°, while 0.75% ZIF-71 NP-incorporated PVDF-HFP ENMs showed increased hydrophobic properties with a WCA of 135°, and enhanced chemical and mechanical stability. The DCMD efficiency while using this nanostructured MD membrane was reported as 99.5% [80].

**Figure 3 membranes-13-00338-f003:**
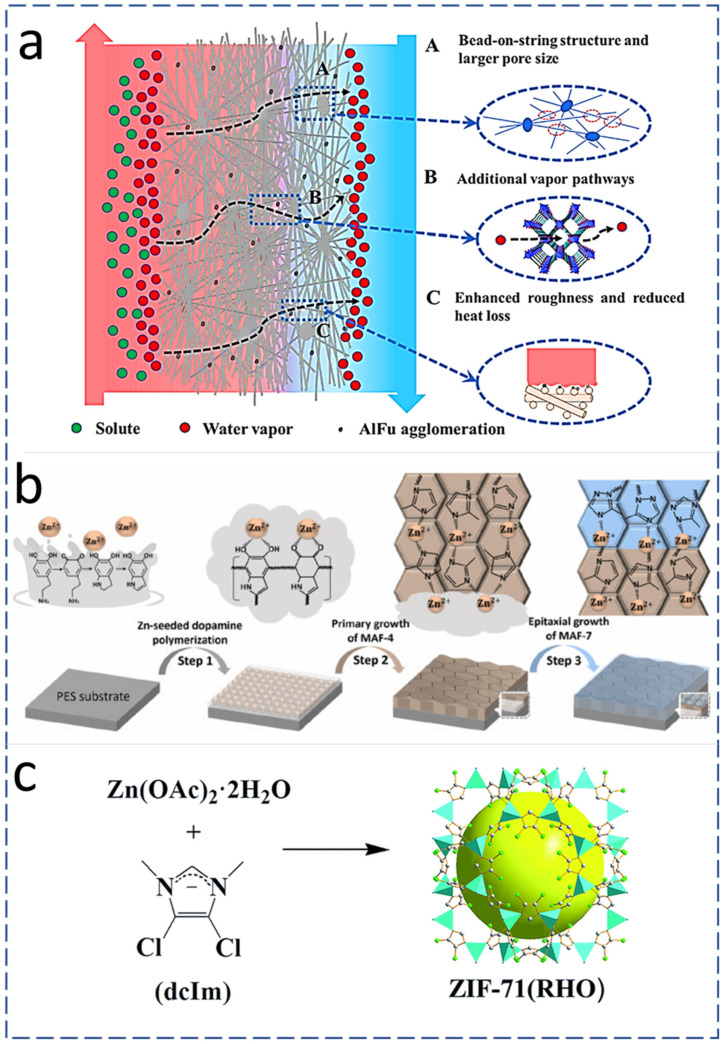
(**a**) Mechanism and effect of AlFu MOF on the DCMD performance [80] (Reprinted with the permission License No: 5487001092782 15 Febuary 2023, Elseviers). (**b**) MAF growth on PES polymer substrate and its conversion to other zeolitic species (MAF-7) [86] (Reprinted with the permission License No: 5490181160662 15 Febuary 2023, Elseviers). (**c**) Crystal structure of ZIF-71: Zn (polyhedral), Cl (green sphere), N (purple sphere), C (white sphere) [89] (Reprinted with the permission License No: 1324377-1 15 Febuary 2023, RSC Publishing).

**Table 2 membranes-13-00338-t002:** ENMs incorporated with MOFs.

ENMs	MOF Incorporated	Content %	ΔT(DCMD) (°C)	Improved Flux %	Reference
Tri-layer PVDF-PAN-PVDF	Hydrophobic SiO_2_/MOF/ hydrophilic SiO_2_	5/1.5/1	30	64.2	[58]
PVDF	MOF (iron 1,3,5-benzenetricarboxylate)	5	32	56.8	[84]
PVDF-HFP	AlFu MOF	2	40	114	[80]
PVDF	MAF-4	8%	40	60%	[88]
PVDF-HFP	ZIF-71	0.5%	40	30	[91]

### 4.2. Effect of SiO_2_, TiO_2,_ and Zinc Oxide (ZnO) on ENMs and the MD Process

Nanomaterial additives such as SiO_2_, TiO_2_, and ZnO can be directly blended with the electrospinning dope solution, and this is a popular approach to obtain modified ENMs for the MD process [59,93]. ENMs incorporated with the aforementioned nanomaterials possess enhanced MD membrane characteristics such as WCA, LEP, mechanical strength, narrow pore size distribution, and controlled porosity [93,94,95,96].

Table 3 shows the MD membrane characteristics and calculated MD flux while using ENMs incorporated with SiO_2_, TiO_2_, and ZnO nanomaterials. In general, these NPs have been utilized for various applications such as biomedical, environmental, textile, and water treatment [97], and their details are discussed in the following section.

#### 4.2.1. SiO_2_ Addition

SiO_2_ NPs have been utilized for various applications because of their size range between 5 to 1000 nm, high adsorption capacity, high specific surface area, unique optical properties, low density, low toxicity level, and biocompatibility. As per the report, the LEP of ENMs with 1% SiO2 was 43% higher than the neat PVDF, as given in Table 3. It may be because the presence of additional nanomaterials on the membrane surface could reduce the pore size, enhancing the surface roughness and thereby the hydrophobicity and WCA [98].

Compared with neat hydrophobic PVDF ENMs with a WCA of 92.8°, PVDF ENMs with modified SiO_2_ NPs showed increased hydrophobicity with a WCA of 109°. Similar WCA has been reported for PVDF membranes with a 3–4% addition of SiO_2_ NPs [99]. The advantage of superhydrophobicity and the “lotus effect” is not limited to its anti-wetting characteristics, but also applies to the membrane’s anti-fouling properties and self-cleaning characteristics [100].

At a temperature of 60 °C, the MD water flux while using SiO_2_-incorporated PVDF ENM (19.4 L m^−2^ h^−1^) was 43% higher than that of neat PVDF ENM (13.6 L m^−2^ h^−1^). Water flux enhancements are also attributed to reduced pore blockage due to the reduced salt deposition or scaling propensity while employing SiO_2_ NP-modified ENMs during the MD process testing [101,102]. In one report, modified SiO_2_ NPs were incorporated into a PVDF polymeric matrix to obtain nanocomposite-based PVDF ENMs with a WCA as high as 147.8–161.2°, which was up to a 10.7% increase compared with the pristine PVDF [93]. Silanes are monomeric silicon compounds such as N-octadecyltrichlorosilane (ODTS), octadecyltrimethoxysilane (OTMS), chloromethyl-octadecyl silane (Cl-DMOS), etc., which are useful for improving the hydrophobic properties of ENMs by lowering their surface energy. The OTMS is an aliphatic long chain of carbon (CH_3_(CH_2_)_17_−) with (-Si-OCH_3_)_3_ as an anchor group [103], whereas Cl-DMOS and ODTS possess (-Si-Cl_3_)_3_ and (-Si-ClCH_2_)_3,_ respectively, as the anchor groups [61,104]. These silanes have plenty of non-polar CH_3_ groups which impart hydrophobic characteristics. Apart from these non-polar groups, the strong electron-withdrawing atoms, viz., oxygen and chlorine in ODTS and Cl-DMOS, result in uneven electron distribution with minimum polarity at the respective sites, which ultimately increases the hydrophobic characteristics [105,106].

The superhydrophobic features may result in the “lotus effect” and thereby yield self-cleaning characteristics to the modified ENMs [100]. To achieve this effect, they must have a WCA close to 180° and a relatively lower sliding angle which may allow for the roll-off of water droplets. These properties would enhance the anti-fouling and anti-scaling characteristics of the membrane [97]. Several other studies have reported increased hydrophobic properties of MD membranes by the addition of SiO_2_ NPs and TiO_2_ NPs. These nanoparticles cause surface roughness, reduce the pore size, and impart respective functional groups to the surface to inhibit foulants or scalants during the MD process. The surface roughness results in increased WCA, which improves the hydrophobic characteristics of the ENM. This hydrophobic behavior creates air pockets that resist pore wetting and let the water droplets easily roll off over the membrane surface [4,97].

#### 4.2.2. TiO_2_ Addition

TiO_2_ is one of the key ingredients that has been utilized to fabricate composite membranes for different applications. TiO_2_ NPs show different behavior with regard to their potential affinity towards H_2_O or any waste effluents present in the water. TiO_2_ NPs are multifunctional due to their versatile characteristics such as large specific surface area and easily tunable chemical properties [107]. The hierarchical structure morphology can be generated by the deposition of TiO_2_ NPs using different tuning agents with desired surface properties. Additionally, TiO_2_ might offer reactive sites for covalent bonding between hydrolyzed silane coupling agents and hydroxyl groups (OH) available on the TiO_2_ surface, as shown in Figure 4a,b.

As shown in Table 3, the ENMs incorporated with TiO_2_ NP (38.71 L m^−2^ h^−1^) showed 5% and 45% enhanced initial flux compared with neat PVDF-HFP (36.78 L m^−2^ h^−1^) and commercial membranes (26.64 L m^−2^ h^−1^), respectively. The reason behind this is the optimum pore sizes and higher porosities of neat PVDF-HFP and ENMs incorporated with TiO_2_, compared to the commercial membranes.

The PVDF-HFP ENMs showed a 5% more porous structure (82.6% porosity) compared with ENMs embedded with the organically modified SiO_2_ NPs (78.5% porosity). This slight decline in porosity was associated with the sprayed NPs on the surface of ENMs that somehow blocked the membrane pores. The PVDF-HFP exhibited a WCA of 142°, which is higher than the commercial membrane with a WCA of 118°. This is because of the rougher surface of the randomly deposited non-woven PVDF-HFP ENM. In one report, the authors compared the WCA of PVDF-HFP with dual layer membranes, which showed a 13°-enhanced WCA compared with one with modified TiO_2_, which had a WCA of 155° and low sliding angles of less than 20°. Moreover, the prepared superhydrophobic dual-layer membranes showed comparable LEP values compared to the commercial membranes [108].

Superhydrophobization can be performed by coating fluorosilane molecules onto TiO_2_ NPs by a low-temperature hydrothermal process. One of the fluorosilanization agents is 1H, 1H, 2H, 2H-perfluorododecyl trichlorosilane (FTCS), which offers sites of hydroxyl groups in TiO_2_ NPs for fluorosilanization and thereby achieves a hierarchical morphology. Hydrophobically modified TiO_2_ NPs can either blend with a polymeric dope solution to fabricate ENMs or disperse in a suitable medium and be coated onto ENMs by electrospraying [107]. The mechanism proposed in Figure 4a reveals the fluorosilanization of TiO_2_ on the surface of PVDF [107]. FTCS anchors hydrophilic trichlorosilane with chains of hydrophobic fluorinated carbon, which reveals the formation of trisilanols/hydrosilanes with hydrolyzation of the hydrophilic trichlorosilane head in water [109]. Further, Figure 4b shows the Si–O–Ti covalent bond formed because of the hydroxyl groups present in trisilanol molecules. These interactions between TiO_2_ and PVDF might offer robust, dense, hydrophobic surfaces which can be beneficial for the MD process [107].

**Figure 4 membranes-13-00338-f004:**
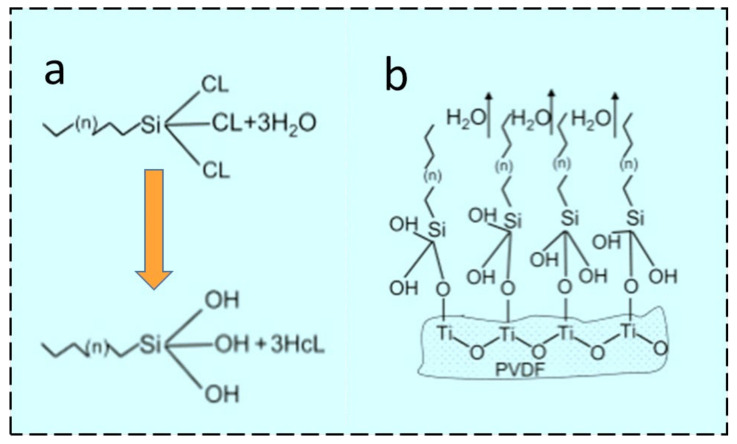
Scheme for the salinization of PVDF by (**a**) hydrolyzation using FTCS, or (**b**) condensation of trisilanols [107] (Reprinted with the permission License No: 5497580436300 14 Febuary 2023, Elseviers).

#### 4.2.3. ZnO Addition

ZnO NPs possess good thermal resistance, high surface-to-volume ratio, and anti-bacterial characteristics. In addition, these nanomaterials are considered environmentally friendlier and more economical than TiO_2_ and Al_2_O_3_ when utilized for surface modification of membranes. ZnO may offer anti-wetting, anti-fouling, and anti-scaling characteristics with reduced water sliding angles when treated with organo-silane molecules such as 1H, 1H, 2H, 2H-perfluorooctyltriethoxysilane (FAS). These features of ZnO with FAS offer a stable superhydrophobic surface, which directly indicates its potential to be utilized for MD [110].

PVDF-HFP with 25% ZnO NPs can be used to fabricate nanostructured ENMs. A stable DCMD flux rate was observed while using a low surface tension feed, with a salt rejection of 99.99% for up to 80 h of continuous operation. A slight compromise in water flux (up to 7% reduction) was observed when compared with neat PVDF-HFP ENMs. The WCA of ZnO NPs with PVDF-HFP ENMs was observed to be as high as 161°, and the average pore size was 0.6 μm, which ultimately impacted the LEP of membranes and reached 187 kPa. This report confirms the potential of ZnO utilization for MD applications [110].

**Table 3 membranes-13-00338-t003:** Characteristics and MD flux of ENMs incorporated with SiO_2_, TiO_2_, and zeolite for MD.

ENMs	Optimized Nanomaterial Concentration	Water Flux	WCA	Pore Size (μm)	LEP (kPa)	Salt Rejection	Reference
PVDF-SiO_2_	1%	19.4 L m^−2^ h^−1^	~109°	~1.48	~64	99.99%	[96]
PVDF-SiO_2_	8%	25.7 Kg m^−2^ h^−1^	~152°	~0.2	~164	99.99%	[111]
PVDF-HFP/PS/SiO_2_	6%	28.1 L m^−2^ h^−1^	~156°	~1.5	~136	100%	[112]
PVDF-HFP/TiO_2_	2.8%	38.7 L m^−2^ h^−1^	~155°	~0.7	~105	99.99%	[31]
PVDF-HFP/ZnO	25%	22.7 L m^−2^ h^−1^	~161°	~0.6	~187	99.99%	[110]

### 4.3. Effect of CNT, GO, and AC on the MD Membranes and the MD Process

Carbon-based hydrophobic nanomaterials, viz., CNTs, graphene, and activated carbon (AC), are being widely utilized as additives in ENMs to fabricate MD membranes with desired characteristics [81,113,114].

#### 4.3.1. CNT Addition

Recently, researchers have optimized the concentration of CNTs in a PVDF-HFP polymeric matrix to obtain highly hydrophobic and robust MD membranes. A 0.5 wt% CNT–PVDF-HFP heat-pressed at 150 °C (CNT-150) yields a robust MD membrane with high DCMD permeate flux (16.5–18.5 L m^−2^ h^−1^), which is 42–50 % higher flux compared with commercially available membranes (11–13 L m^−2^ h^−1^) at similar operating conditions. ENMs generally show higher MD vapor flux than commercial membranes, which is primarily attributed to their unique porous structure and surface roughness as shown in Figure 5a. The porosity of untreated PVDF-HFP ENMs was 89%, while the membrane after heat-press (150 °C) treatment resulted in a decreased porosity (80%) because of the reduction in certain voids and pore size after heat treatment. Conversely, the stress at break and Young’s modulus of the treated ENM was increased due to the compaction of fibers after heat-press treatment. The WCA of PVDF-HFP ENMs was reduced from 135.9° to 123° because of the reduction in surface roughness as a result of the heat-press treatment and the melting of partial nanofibers. Thus, a controlled heat treatment yields robust ENMs with a slight reduction in the porosity and WCA. Additionally, 99.99% of inorganic non-volatile salt rejection can be achieved by using CNT-incorporated electrospun MD membranes during a DCMD process as mentioned in Table 4 [81].

#### 4.3.2. GO Addition

Reduced graphene oxide (rGO) is hydrophobic, and it is considered a promising candidate for ENM-based nanocomposite MD membranes [82,115]. Fine and porous membranes based on GO exhibit a unique permeation pathway for water molecules. The rGO-ENMs have very good mechanical properties, chemical stability, flexibility, anti-fouling properties, and hydrophilicity [116]. The physicochemical comparison of GO and rGO is shown in Figure 5b. GO contains -O-, –OH, C=O, and –COOH groups, whereas rGO is a reduced form of GO with reduced hydrophilic characteristics, which is required for the MD process.

The hydrophilicity of GO originates from the hydroxyl and epoxy groups in the basal plane, along with other functional groups such as carbonyl and carboxylic groups [117]. Octadecyl groups are substituted by GO groups upon functionalization [108], which results in an active surface area, higher hydrophobicity, higher surface contact angle, and lower surface energy. In a recent report, a commonly used three-stage method was used to fabricate superhydrophobic mixed-matrix PVDF-HFP ENMs that were modified using octadecylamine-reduced graphene oxide (ODA-rGO) and utilized for the MD process. The resultant PVDF-HFP ENMs with ODA-rGO showed a superhydrophobic nature with a WCA of 162°. The contact angles of the superhydrophobic nanofiber membranes containing 0.1 wt% and 0.5 wt% ODA-rGO were measured as 158° ± 1° and 156° ± 3°, respectively, which were about 14.5% and 13% higher than the contact angle of the pristine PVDF-HFP, as shown in Table 4. Beads-on-string morphology was observed due to the agglomeration of ODA-rGO (0.5 wt%) during the elongation of the nanofibers. In addition, the fiber diameter could be tuned by changing the concentration of ODA-rGO in the dope solution [118].

Additionally, LiCl (0.005 wt%) was used to control the pore size of ENMs followed by a hot-pressing process. The average pore size of ENMs was reduced from 1.30 µm to 0.24 µm, and the LEP was improved from 30.4 kPa to 127.6 kPa in comparison with the pristine ENMs upon heat pressing, as shown in Table 4. The tensile strength of hot-pressed rGO-ENMs was increased by about 26% compared with the neat ENMs [119].

#### 4.3.3. AC Addition

Nanostructured AC is widely utilized for water treatment applications because of the various available functional groups on its surface, which are beneficial for modifying the surface and its physical and chemical characteristics [83]. In a recent report, the effect of AC NP-based ENM’s hydrophobicity was studied, which revealed that the AC NPs form a hierarchical structure with micro-wrinkles or protrusions and results in increased surface roughness, ultimately increasing the surface hydrophobic properties. This is because of the result of weave-like surface roughness generated on the ENMs. This results in increased WCA for all samples incorporated with AC. Figure 5c shows a schematic illustration of an AC-anchored nanofiber structure [64]. Recently, dual-layer compositions with hydrophobic and hydrophilic layered membranes were prepared and utilized for the DCMD process.

The selective hydrophobic PVDF-HFP ENMs were prepared with varying AC NP concentrations from 0 to 3.0 wt% along with other hydrophilic support layers [64]. Overall MD performance was improved with the composition of 1.5 wt% NPs, yielding a water vapor flux of 45.6 L/m^2^ h which is 9% higher compared with the MD flux while using commercially available PTFE membranes (41.8 L/m^2^ h) with no compromise in the salt rejection (99.99%) during the MD process. These resultant membranes have relatively wider pores, which contributed to an increase in the porosity by ~20% compared with the commercial PTFE membrane. The higher porosity of these fabricated ENMs is beneficial for reducing mass transfer resistance and leads to better permeation flux during the MD process. The ENMs exhibit relatively lower LEP values (1.21–1.36 bar) than the commercial PTFE membrane due to their larger pore size. However, the LEP values of these ENMs are quite comparable with other results obtained for MD membranes reported in the literature [76,120,121,122,123], which is considered to be adequate for DCMD applications.

L-type and H-type are two kinds of ACs. H-type ACs are positively charged when treated under water or strong acids and thus possess a hydrophobic character. On the other hand, L-type ACs are neutralized by strong bases and possess hydrophilic properties [124]. Typical oxygen-containing surface functional groups in AC are depicted in Figure 5d, and these oxygen-containing surface functionalities are the routes for achieving desired surface properties during membrane fabrication [125].

**Figure 5 membranes-13-00338-f005:**
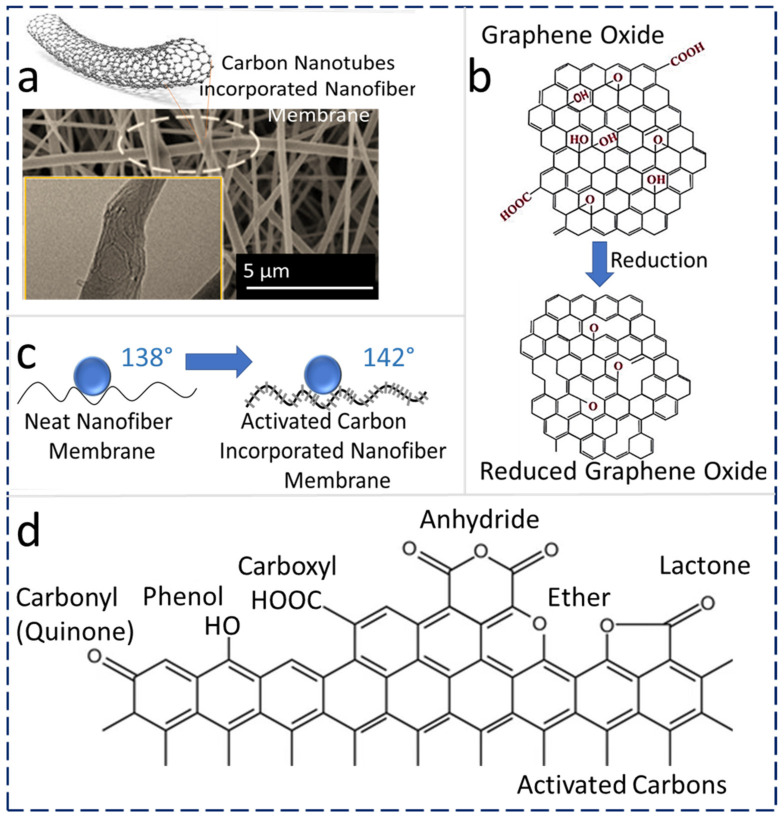
(**a**) CNT-incorporated nanofibers and structure [114] (Reprinted with the permission License No: 5475271453015 24 January 2023, Elseviers). (**b**) Conversion of GO to rGO [126]. (**c**) Protrusion effect on WCA on ENMs produced by AC (**d**) Chemical structure of ACs.

**Table 4 membranes-13-00338-t004:** Physical properties of ENMs incorporated with CNTs, rGO, and AC for MD.

ENMs	Guest Material	Optimized Concentration	Water Flux	WCA	Porosity (%)	Strength (Mpa)	LEP (kPa)	Salt Rejection	Reference
PVDF-HFP	Carbon nanotubes	0.5%	19.2 L m^−2^ h^−1^	140.7 ± 2.2°	87 ± 2.5	52.09 ± 0.75	50 ± 2.0	99.99%	[114]
PVDF-HFP	ODA-rGO	0.1%	21.1 kg m^−2^ h^−1^	158° ± 1°	70.5 ± 0.3	20.94 ± 5.60	127.6 ± 1.2	99.99%	[119]
PVDF-HFP	Activated carbon	1.5%	45.6 L m^−2^ h^−1^	142.7 ± 0.6°	90.5 ± 1.7	17.8	136 ± 4	99.99%	[64]

## 5. Wetting, Fouling, and Scaling Behavior of Modified Electrospun MD Membranes

Pore wetting, membrane fouling, and scaling are technical challenges in MD process that reduce the efficiency of the process and the shelf life of the membrane. Horseman and co-workers have described state-of-the-art insights on the fundamental mechanisms and mitigation strategies for wetting, scaling, and fouling in the MD process. Practical MD processes often involve complex feed solutions and the simultaneous occurrence of wetting, scaling, or fouling phenomena. As per this report, most of the studies separately focus either on a single mechanism (wetting/fouling/scaling) or multiple mechanisms. They emphasize the importance of mitigation strategies in the MD process by understanding the fundamental and complex mechanisms of fouling and scaling. As per their recommendation, the MD process study should be aim at pretreatment, operation, and membrane development to address the multiple failure mechanisms when using real feed solutions with more complex foulants and scalants [127].

There are many reports on the fabrication of fouling- or scaling-resistant hollow-fiber and flat-sheet MD membranes. Many efforts from the researchers are reported to mitigate the temperature polarization (TP) or concentration polarization (CP) phenomenon and thereby mitigate the wetting and scaling issues in the MD process. Some of the polarization mitigation techniques are: employing spacers, bubbling or micro-bubbling, patterned membranes, flashing the feed, localized heating, etc. [128,129,130,131,132,133,134].

Kim et al. reported a flux enhancement as high as 43% by introducing feed and permeate spacers with a hydrodynamic angle of 90° during a DCMD process test. A study on the introduction of innovative swirling-flow microbubbles into a DCMD module resulted in a 37% flux enhancement. An 87% performance enhancement was reported during a DCMD process while using a corrugated membrane. Flashing the feed solution in a VND process yielded a 3.5-fold increase in the water permeate flux by mitigating the TP phenomenon. A report on the introduction of localized heating in an MD module demonstrated a reduction in heat loss and the TP effect [135]. Most of these mitigation strategies were not employed using electrospun MD membranes, and there is a research gap in this area of understanding the heat and mass transport mechanisms and MD process performance. Recently, researchers have become more interested in the fabrication of modified electrospun MD membranes with anti-scaling and anti-fouling characteristics by incorporating nanomaterials or by chemical modifications [10,77,136,137,138,139,140,141]. Table 5 shows a variety of electrospun MD membranes having anti-fouling/anti-scaling properties.

From the above table, most of the reported works use PVDF electrospun membranes modified with nanoparticles such as silica, silver, POSS, graphene oxide, and carbon nanotubes. The presence of nanoparticles in the electrospun membranes enhances membrane characteristics such as mechanical strength, LEP, and water contact angle along with the anti-wetting, anti-fouling and/or anti-scaling properties of the membrane. Most of the reported works demonstrate the blending of the nanomaterials along with the electrospinning dope solution before the membrane fabrication [147,154].

In the electrospun membranes obtained from these blended dope solutions, the majority of the nanomaterials are distributed in the bulk rather than on the membrane surface. So a minimum amount of used nanomaterials would be available to impart anti-fouling or anti-scaling properties to the membrane. Instead of blending the nanomaterials with the polymeric dope solution, if the nanomaterials can coat the membrane surface in the second step in an efficient way, most of them can be available in the antibacterial or anti-scaling action during the water-treatment process. Dispersion of a homogeneous nanomaterial solution on the membrane surface by electrospraying is recommended as a robust method for fabricating nanocomposite membranes [147].

Nanostructured composite membranes with other possible hydrophobic electrospun non-woven substrates [155,156] and other nanomaterials having anti-bacterial or anti-fouling properties have to be explored for MD applications. Detailed failure mechanisms due to multiple fouling or scaling phenomena have to be explored with the help of experimental investigation and computational fluid dynamics. Membrane-integrated 3D spacers could be another area to be explored to mitigate membrane fouling and scaling propensity [157,158].

## 6. Conclusions and Future Outlook

This review article emphasizes the importance of tuning MD membrane characteristics by incorporating the optimum quantity of nanostructured materials into ENMs. Variable tuning parameters that have to be kept in mind during an electrospinning process to fabricate defectless ENMs are summarized. Enhancements in the MD membrane characteristics upon the addition of nanomaterials such as MOFs, zeolites, SiO_2_, TiO_2_, ZnO, CNT, GO, and AC are discussed. MD process performance based on water vapor flux and developments in the fabrication of anti-wetting, anti-fouling, and anti-scaling ENMs by incorporating various nanofillers are also discussed in this article. ENMs are comparatively economical when a small number of NPs are incorporated into a polymer solution for electrospinning; however, the optimization of several parameters, optimum selection of NPs, and the need for well-planned studies still need to be addressed [78].

Electrospinning is a versatile approach for membrane fabrication but there are still many improvements to be made. Electrospinning can be directly performed on different types of collectors and solutions with alcohol as a non-solvent or on sonicated solutions, which may directly react faster because of their finer diameter. These approaches may offer the direct formation of superhydrophobic ENMs with special morphologies, ultimately yielding high flux and better MD performance with self-cleaning membrane characteristics [97]. Recycled RO membranes, RO feed spacers and RO permeate spacers from discarded RO membrane modules have been used as ENM supports for the MD process and compared to the performance by Jorge et al. [159]. ENMs fabricated from recycled plastics have not yet been employed for MD process testing. Similarly, various polymers with low surface energy are available but have not yet been investigated for MD. A variety of NPs and surface functionalization have been reported for other applications, but there is a big research gap to be explored in their incorporation or respective chemical treatments of ENMs for MD [153]. Modified ENMs with chemical vapor deposition (CVD), physical vapor deposition (PVD) and plasma treatment for enhancing MD membrane characteristics have to be explored. Advanced MD configurations also need to be explored with novel pristine ENMs and blended composite and modified MD membranes prepared via an electrospinning process [61]. Electrohydrodynamic atomization of nanomaterials to coat membrane surfaces to tune the surface characteristics is rarely explored [160]. So far, the application of ENMs in the MD process has been limited to lab-scale investigations. Pilot-scale and large-scale electrospun MD membranes have to be fabricated and tested. Apart from that, more scientific modeling and simulation tools are very important to understand the heat and mass transport mechanisms during the MD process and to validate the experimental data. These tools will also help to up-scale the lab-scale membrane module and process to a large-scale system.

## Figures and Tables

**Figure 1 membranes-13-00338-f001:**
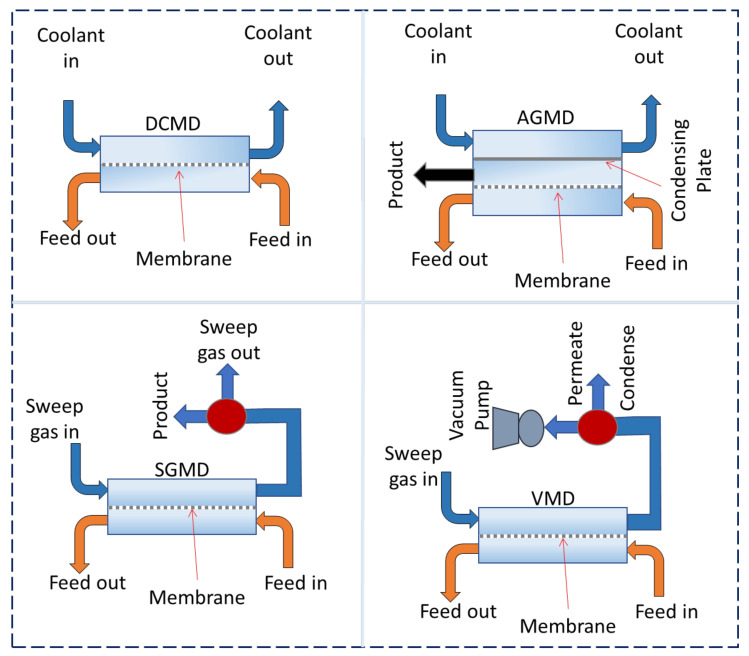
Schematic view of different MD processes: DCMD, AGMD, SGMD, and VMD [40].

**Figure 2 membranes-13-00338-f002:**
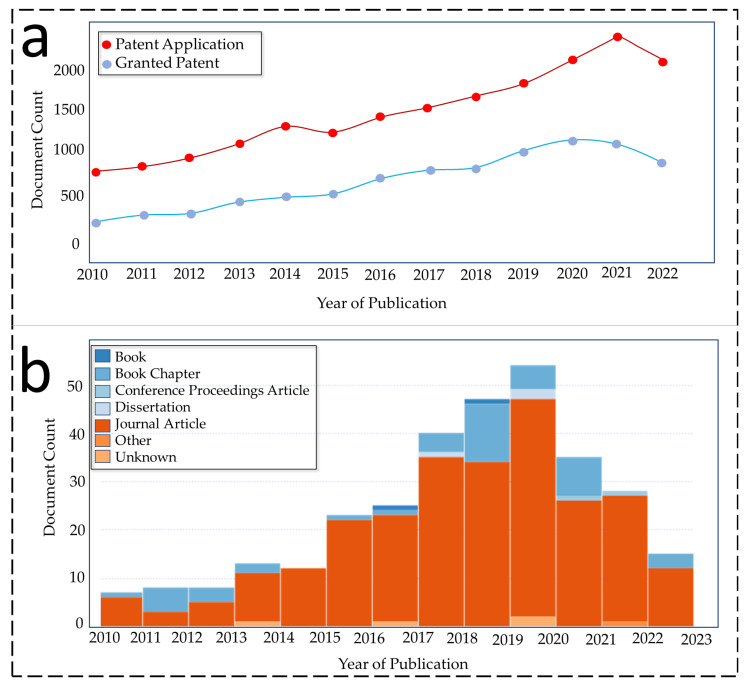
(**a**) Patent records found on www.lens.org (accessed on 20 January 2023) with keywords “nanofibers for membrane distillation”. (**b**) Scholarly records found on www.lens.org (accessed on 20 January 2023) with keywords “nanofibers for membrane distillation, since 2010”.

**Table 1 membranes-13-00338-t001:** Effect of electrospinning parameters on electrospinning process and respective advantages.

Electrospinning Parameter	Effects on the Electrospinning Process	Remarks	Reference
Polymer Type/Solvent	Suitable polymers and solvents are selected for respective polymeric solutions used for electrospinning. Optimization of viscosity, concentration, conductivity, and surface tension are required to form bead-free or beaded nanofibers as per required membrane characteristics.	The chemical and physical properties of resultant ENMs can roughly be predicted by considering the properties of the polymer solution which is intended to be utilized for electrospinning.	[65]
Molecular Weight of Polymer	The molecular weight of the polymer affects the resultant electrospinning polymer solution properties which can be optimized by the addition of acid/salt/suitable guest molecules to obtain the desired morphology of resultant nanofibers.	The rheology of electrospinning polymer solution is directly attributed to the molecular weight, which is advantageous for electrospinning optimization with regard to the surface properties of membranes.	[66]
Polymer Concentration	Polymer concentration is the key factor to set viscosity and conductivity for a smooth electrospinning process, where synchronization between the voltage supply and polymer concentration is very important.	Concentration may result in the formation of fibers with small to large average diameters. The pore size of the ENMs can be optimized by tuning the fiber diameter and membrane thickness.	[23,24]
Conductivity	The conductivity of the electrospinning polymer solution affects the formation of the Taylor cone. Increased conductivity may form finer fibers at low voltage supply or cause tip blockage due to charge accumulation.	The dope solution with higher conductivity needs only a low-voltage supply. Comparatively finer fibers can be formed when a solution with higher conductivity is utilized for electrospinning.	[24,67]
Surface Tension	Surface tension affects the proper fabrication of beaded or bead-free nanofibers.	Depending on the end-use, smooth fibers or fibers with beads can be prepared by varying surface tension. Beaded or beadless surface morphology has an impact on their hydrophobicity or hydrophilicity.	[53]
Viscosity	Increased polymeric concentration will increase the viscosity, and maximum/minimum viscosity needs to be optimized.	Maximum viscosity of solution regardless of bead formation will be beneficial.	[68]
Voltage Supply	The voltage supply normally used for electrospinning is about 10–30 kV, which sets the speed of fibers coming out of the tip during electrospinning.	The greater the voltage supplied, the higher will be the concentration of electrons pushing the polymer solution toward the collector frequently. Optimized voltage will be helpful to reduce the cost of resultant ENMs and reduce the chances of tip blockage and formation of polymer waste.	[66,69]
Tip to collector distance	Increased tip-collector distance plays an important role in stretching the nanofiber and making nanofibers finer.	Depending on the end-use, the tip-to-collector distance can be increased/decreased to get fibers with the desired average diameter	[69]
Humidity	Increased humidity may form a circular porous network in fine electrospun nanofibers and may lead to saturated porosity; thus, optimization is needed depending on the end-use. It also resists the solidification of polymer solutions and controls the consistency of the electrospinning process.	Optimum humidity offers smooth electrospinning. This will resist tip blockage while electrospinning and may offer frequency of fiber diameters in a smaller range with a uniform morphology.	[68,70]
Temperature	The rate of solvent evaporation is directly proportional to an increased temperature during electrospinning, which may affect the average thickness of fibers produced.	For less volatile or non-volatile solvents, temperature plays a key role in running a smooth electrospinning process. In addition, the thickness of nanofibers can be tuned by varying the temperature to get ENMs with desired characteristics.	[71]
Feed rate	Feed rate optimization is important to achieve smooth electrospinning with beads or bead-free electrospun fibers. In most cases, the optimum feed rate is 0.5–2.0 mL per hour.	Feed rate is an important parameter which to resist tip blockage during the electrospinning process. Optimized feed rate may offer uniform beaded/bead-free fiber morphology and fibers with desired diameter.	[65,72]

**Table 5 membranes-13-00338-t005:** Modification of electrospun membranes for gaining anti-fouling or anti-scaling properties.

Membrane	Modification	Remarks	Configuration	Reference
Polyimide	Organo-silica mesoporous POSS	Fouling-resistant MD membrane with enhanced flux	DCMD	[142,143]
PVDF	Electrospraying of PVDF/PDMS/Silica	Robust and superhydrophobic membranes with anti-fouling and anti-scaling properties	DCMD	[141]
PVDF	Zwitterionic bilayer membrane	Produced water treatment with anti-wetting, and anti-fouling characteristics	DCMD	[144]
PVDF	Silica + AgNP + carbon nanotubes	Membrane with superhydrophobic, anti-fouling, and anti-wetting properties	DCMD VMD	[99,145]
PVDF	Mixed matrix with carbon-based fillers	Dual- and triple-layer superhydrophobic membranes with anti-wetting properties	DCMD	[146]
PVDF-HFP	Electrospraying of carbon nanotubes	Membranes with anti-scaling properties and enhanced TPC	DCMD	[147]
PVDF	Hyperbranched dendritic structure with nitrilotriacetic acid	Stable flux and anti-fouling characteristics	AGMD	[148]
PVDF	POSS functionalized graphene oxide	Anti-fouling membranes for arsenic removal	AGMD	[149]
PVDF-HFP	Functionalized POSS	Amphiphobic, anti-surfactant-wetting membrane	DCMD	[150]
Styrene–Butadiene–Styrene	Not applicable	Elastomeric membrane with anti-scaling and anti-fouling properties	DCMD	[151,152]
PVDF	Silane—chemical treatment	Anti-fouling and anti-wetting MD membranes	DCMD	[153]
PVDF	Silica NP	Three-dimensional superhydrophobic wetting-resistance ENMs with improved flux	DCMD	[111]
PVDF-HFP	ZnO NP	Dual-layered robust membranes with anti-wetting and anti-scaling characteristics	DCMD	[110]
PVDF	Silica NP	Robust oil-fouling resistant and anti-wetting ENMs	DCMD	[112]

## Data Availability

Not applicable.

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
