# Peer review of "Modified Electrospun Membranes Using Different Nanomaterials for Membrane Distillation"

_membranes, 2023, doi:10.3390/membranes13030338_

Round 1

Reviewer 1 Report

1. The writing style needs to be reconsidered. Some paragraphs are too short and fragmented. Besides, there are some expressions that should be further improved. Some parts are also poorly written. 

2. There are some editorial problems. (i.e. line space, space between paragraphs, etc.)

3. "2. Conventionally applied MD configurations": This section is simply redundant. 

4. "3. New developments in nanofiber-based membranes": This section should be rewritten to make its key points clear. Currently, it does not have a clear focus. 

5. "Table 2": More references should be cited.

6. "Table 3": More references should be cited.

7. "Table 4": More references should be cited.

8. "Table 5": More references should be cited.

9. "Conclusions": They are rather general. Moreover, there should be clear insights into future prospects of this topic. 

Reviewer 2 Report

This manuscript introduces how to adjust the characteristics of membranes by adding an appropriate number of nanostructured materials. In the process of electrospinning, in order to improve the ENM, some parameters must be adjusted. The enhancement effect of nanomaterials (such as MOFs, SiO2, TiO2, zeolite, carbon nanotubes, graphene/oxide, and AC) on the characteristics of MD membrane was discussed. This manuscript also discusses the MD process performance based on water vapor flux, and the research progress of anti-moisture, anti-fouling, and anti-scaling ENM prepared with different nano-filler.

Here are some issues this reviewer's major concern:

1. In general, the tables in this manuscript are more than the relevant model figures and representation figures. Some contents are difficult to understand and do not give readers an intuitive impression. The summary is too brief, which makes some theories difficult to understand. It is suggested to improve and enrich the basis of this manuscript.

2. In section 4.1, the "leaching" problem raised is not introduced. The understanding of three-dimensional porous structures based on AlFu's MOF is relatively abstract, and it is suggested to add relevant structural figures.

3. In line 267, the theory of "available electrons attract strong electrons" has been quoted many times in the manuscript, but all of them have passed by. It is strongly suggested that the theory of multiple applications can be introduced in detail and relevant model diagrams can be added.

4. In line 277, it is mentioned that two factors in other studies increase hydrophobicity, but it does not specify how to increase it. It is suggested to supplement it.

5. In section 4.2.2, it is recommended to introduce the chemical properties that TiO2 can use in detail. The readers have a vague understanding of "easily adjustable chemical properties”.

6. In section 4.2.3, the author briefly mentions the different skeleton diagrams of zeolite and MOF. It is suggested to attach the structural diagrams of the two to make readers better understand and have a deeper impression.

7. In section 4.3.3, it is recommended to introduce the chemical properties of AC that can be applied to MD in more detail and construct the structure diagram of AC NPs to enrich this section.

8. In the section on membrane fouling, this manuscript briefly mentions the technologies used to mitigate membrane fouling and membrane wetting in MD but does not introduce them in detail. It is suggested to enrich this section more.

Reviewer 3 Report

This is a mini-review on electrospun membranes for MD in the passed ten years. It is some useful for readers. But is should be carefully revised before accept. And some questions are as follows.

1)       The title “Engineered Electrospun Membranes for Membrane Distillation” is too wide. The mini-review is organized on the addtion of some substances to ENMs (MOF, SiO2, TiO2, Zeolites, CNT, GO, AC). So it should be revised as a specialized title to fit this.

2)       And the word “engineered” is not good here. It is easy to make a confusion for this membrane because design includes more info.

3)       Table 1 should provide some data of the elecrospining parameters and their results, which is good for membrane researchers.

4)       Some writing mistakes include “an MD”, “from Seawater”, “, Electrospun”, “high water contact angle”,” 118 kPa This…” and so on.

Round 2

Reviewer 2 Report

The authors have dressed all my concerns.

Reviewer 3 Report

This version has been revised by the comments.